# Token-by-Token Regeneration and Domain Biases:
# A Benchmark of LLMs on Advanced Mathematical Problem-Solving

**Evgenii Evstafev**[*]
University of Cambridge

## Abstract

Large language models (LLMs) excel in many natural language tasks, yet they struggle with complex mathematical problem-solving, particularly in symbolic reasoning and maintaining consistent output. This study evaluates 10 LLMs with 7 to 8 billion parameters using 945 competition-level problems from the MATH dataset. The focus is on their ability to generate executable Python code as a step in their reasoning process, involving over 9,450 code executions. The research introduces an evaluation framework using mistral-large-2411 to rate answers on a 5-point scale, which helps address inconsistencies in mathematical notation. It also examines the impact of regenerating output token-by-token on refining results. The findings reveal a significant 34.5% performance gap between the top commercial model (gpt-4o-mini, scoring 83.7%) and the least effective open-source model (open-codestral-mamba:v0.1, scoring 49.2%). This disparity is especially noticeable in complex areas like Number Theory. While token-by-token regeneration slightly improved accuracy (+0.8%) for the model llama3.1:8b, it also reduced code execution time by 36.7%, highlighting a trade-off between efficiency and precision. The study also noted a consistent trend where harder problems correlated with lower accuracy across all models. Despite using controlled execution environments, less than 1% of the generated code was unsafe, and 3.17% of problems remained unsolved after 10 attempts, suggesting that hybrid reasoning methods may be beneficial.

## 1    Introduction

Large language models (LLMs) have demonstrated remarkable proficiency in natural language tasks (1), yet their ability to solve complex mathematical problems remains constrained by challenges in symbolic reasoning and precise output formatting (2). While recent advances, such as code-augmented problem-solving, offer promising pathways (3), systematic evaluations of LLMs' mathematical capabilities—particularly across diverse architectures and difficulty levels—remain underexplored (4). This study addresses this gap by benchmarking 10 LLMs (7B–8B parameters) on 945 competition-level mathematics problems from the MATH dataset (4; 5), focusing on their ability to generate executable Python code as a reasoning intermediate. The investigation introduces two contributions:

- A granular evaluation framework using mistral-large-2411 (6) for automated answer scoring, addressing inconsistencies in mathematical notation (e.g., fractions, symbolic constants).

- An empirical analysis of token-by-token regeneration—a dynamic output refinement technique—applied to llama3.1:8b (7) to assess its impact on accuracy and computational efficiency.

---

[*]University Information Services (UIS), University of Cambridge, Roger Needham Building, 7 JJ Thomson Ave, Cambridge CB3 0RB, UK, ee345@cam.ac.uk

## 2 BACKGROUND AND RELATED WORK

Modern LLMs employ diverse strategies for mathematical problem-solving, including chain-of-thought prompting (10), program-aided language models (PAL) (11), and symbolic equation generation. Code-based approaches, where models generate executable programs to derive solutions, have gained traction for their ability to enforce logical rigor and mitigate hallucination (12). However, output variability—such as inconsistent numerical formatting or symbolic representation (e.g., $\pi$ vs. 3.1416)—complicates automated evaluation, necessitating robust scoring mechanism. Key limitations persist:

- Performance degrades nonlinearly with problem complexity, as seen in GPT-4's 23% accuracy drop on Level 5 MATH problems (4).
- Models exhibit uneven proficiency across mathematical subjects, often struggling with combinatorics and modular arithmetic (13).
- Unrestricted code generation introduces vulnerabilities like infinite loops or unsafe system calls, requiring sandboxed execution environments (14).

These challenges underscore the need for standardized evaluation protocols and architectural innovations to enhance reliability. Existing benchmarks like MATH (4) rely on binary correctness scoring, overlooking partial solutions.

## 3 METHODOLOGY

### 3.1 DATASET CREATION

This study uses the MATH dataset, a publicly available collection of 12,500 challenging competition-level mathematics problems sourced from competitions such as AMC 10, AMC 12, and AIME. The dataset, described in the paper "Measuring Mathematical Problem Solving With the MATH Dataset" (4), includes step-by-step solutions and spans diverse mathematical domains.

A stratified subset of 945 problems (15) was curated to ensure balanced representation across 7 mathematical subjects (Algebra, Counting & Probability, Geometry, Intermediate Algebra, Number Theory, Prealgebra, Precalculus) and 5 difficulty levels (Level 1: simplest, Level 5: most complex). Each subject-level combination contains 27 problems, yielding a total of 7 subjects × 5 levels × 27 problems = 945 problems. This structured sampling guarantees diversity in both topic coverage and problem complexity.

To streamline evaluation, the dataset was augmented with a dedicated field containing final numerical answers (without explanations). This design enables direct comparison between model-generated outputs and ground-truth solutions.

### 3.2 MODEL SELECTION

The study evaluates 10 language models of varying architectures and scales (7B–8B parameters), selected for their computational efficiency and diversity in training methodologies:

1. llama3.1:8b (7): General-purpose LLM with strong NLP capabilities.
2. olmo2:7b (16): Open-source model optimized for research reproducibility.
3. codestral-2501 (17): Code-focused model for code generation tasks.
4. gpt-4o-mini-2024-07-18 (8): Compact variant of GPT-4.
5. granite3.1-dense:8b (18): Dense model trained on large-scale datasets.
6. open-codestral-mamba:v0.1 (9): Hybrid architecture combining code and general capabilities.
7. ministral-8b-2410 (19): Lightweight model for on-device applications.
8. gemini-1.5-flash-8b (20): Efficient model with strong task performance.
9. mistral-small-2409 (21): Smaller variant of Mistral's architecture.

10. command-r7b:7b (22): General-purpose conversational model.

Token-by-token regeneration, a technique to refine outputs iteratively, was exclusively tested on llama3.1:8b to isolate its impact. Evaluation Constraints:

- Each model independently solved all 945 problems.

- Models were instructed to "generate Python code that prints the final answer to the console." (23)

- Code Generation Limit: 2 minutes per attempt.

- Execution Timeout: 1 minute per code run to prevent infinite loops.

- Retry Mechanism: Up to 10 attempts per problem, with error messages (e.g., syntax/runtime errors) fed back to the model for iterative refinement.

A restricted execution environment permitted only safe built-in functions (e.g., mathematical operations, control flow structures) to mitigate security risks (14).

## 3.3 EVALUATION METRICS

To address variability in mathematical answer formats (e.g., fractions, symbolic notation like $\pi$ or decimal representations), the correctness of console outputs was evaluated using mistral-large-2411 (6), a high-performance language model. This approach ensures robust and consistent scoring despite differences in output formatting.

For every problem, the console output generated by a model's Python code was independently assessed by mistral-large-2411. The evaluator model was blinded to the source model's identity to eliminate bias.

Answers were scored on a 5-point scale by comparing the generated output to the ground-truth answer from the MATH dataset:

- 5 (Correct): Exact match or mathematically equivalent result.

- 4 (Almost Correct): Minor formatting discrepancies or rounding errors.

- 3 (Partially Correct): Partial solution with significant inaccuracies.

- 2 (Incorrect): Wrong answer but relevant to the problem.

- 1 (Completely Incorrect): Irrelevant or nonsensical output.

The primary metric is the weighted accuracy, calculated as the percentage of answers scoring 4 or 5 across the dataset. Scores $\leq 3$ are treated as incorrect.

The evaluator model received only the ground-truth answer, and generated console output, without metadata about the source model.

This method quantifies solution quality more granularly than binary correctness, enabling future analysis of incremental improvements (e.g., from "partially correct" to "almost correct").

To address variability in mathematical answer formats (e.g., fractions vs. decimals, symbolic notation like $\pi$ vs. numerical approximations), direct string comparison proves insufficient for robust evaluation. Such discrepancies necessitate expert judgment to assess equivalence, particularly for context-dependent representations.

The mistral-large-2411 model was employed as a proxy for domain expertise, evaluating console outputs against ground-truth answers through semantic equivalence rather than syntactic exactness. Its 5-point scoring scale accommodates partial correctness and formatting nuances, mirroring human expert evaluation. This approach avoids the brittleness of exact matching while ensuring systematic, bias-free assessment across diverse problem types and answer styles.

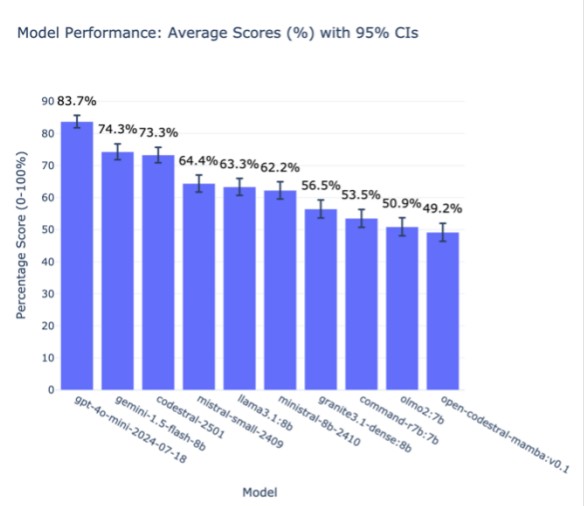

Figure 1: Model Performance – Average Success Rates (%) with 95% Confidence Intervals

## 4 RESULTS

### 4.1 OVERALL MODEL PERFORMANCE

A bar chart comparing the success rates of all models in solving the 945 mathematical problems reveals significant performance disparities. The gpt-4o-mini-2024-07-18 achieved the highest accuracy at 83.7%, while open-codestral-mamba:v0.1 ranked lowest at 49.2%.

### 4.2 PERFORMANCE BY DIFFICULTY LEVEL

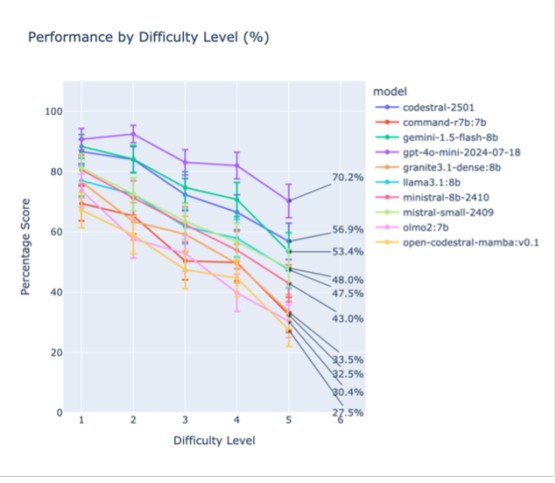

Figure 2: Performance by Difficulty Level (%)

A line graph illustrates a consistent downward trend in accuracy across all models as problem difficulty increases from Level 1 (simplest) to Level 5 (most complex).

This inverse correlation between difficulty and performance highlights persistent challenges in solving advanced mathematical problems.

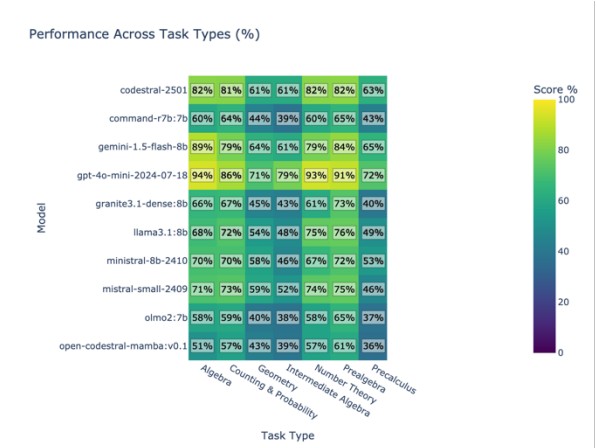

Figure 3: Performance by Task Type (%)

### 4.3 PERFORMANCE ACROSS MATHEMATICAL DOMAINS

A heatmap (Figure 3) visualizes model-specific strengths and weaknesses across seven mathematical domains.

### 4.4 COMPUTATIONAL EFFICIENCY

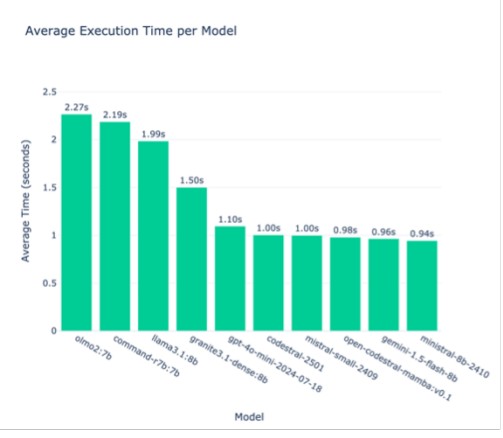

Figure 4: Average Execution Time per Model

A histogram (Figure 4) quantifies the computational efficiency of generated code. ministral-8b-2410 produced the fastest-executing programs (mean: 0.94 seconds), while olmo2:7b generated the slowest code (mean: 2.27 seconds).

### 4.5 IMPACT OF TOKEN-BY-TOKEN REGENERATION ON LLAMA3.1:8B

#### 4.5.1 OVERALL PERFORMANCE COMPARISON

Implementing token-by-token regeneration for llama3.1:8b yielded a marginal improvement in accuracy:

- Original: 63.3%
- Improved: 64.1%

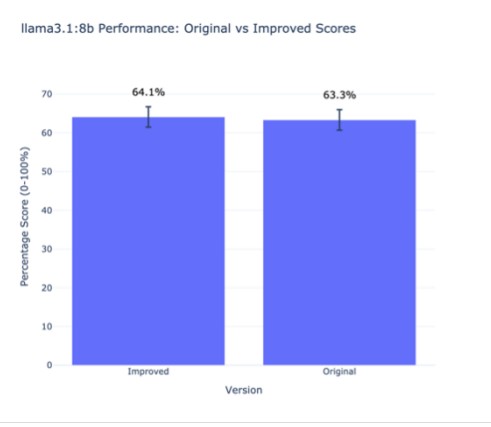

Figure 5: Original vs Improved Scores

This 0.8% gain suggests limited efficacy of the method for general problem-solving (Figure 5).

### 4.5.2 Difficulty-Level Analysis

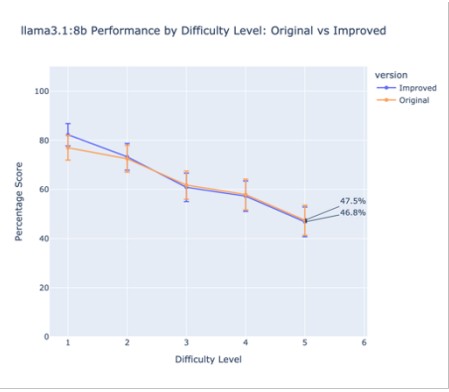

Figure 6: Performance by Difficulty Level – Original vs Improved

Improvements were concentrated in Level 1 problems. Levels 2–5: Statistically insignificant changes. This indicates the technique primarily enhances performance on simpler tasks (Figure 6).

### 4.5.3 Computational Overhead

The improved version reduced code execution time by 36.7%:

- Original: 1.99 seconds
- Improved: 1.26 seconds

Despite minimal accuracy gains, the optimization significantly enhanced computational efficiency.

### 4.5.4 Domain-Specific Effects

Token-by-token regeneration improved Algebra performance but showed neutral or negative effects in other domains.

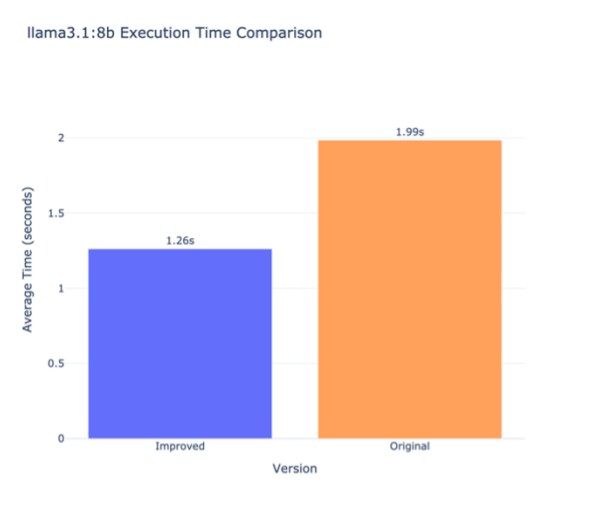

Figure 7: Average Execution Time Comparison

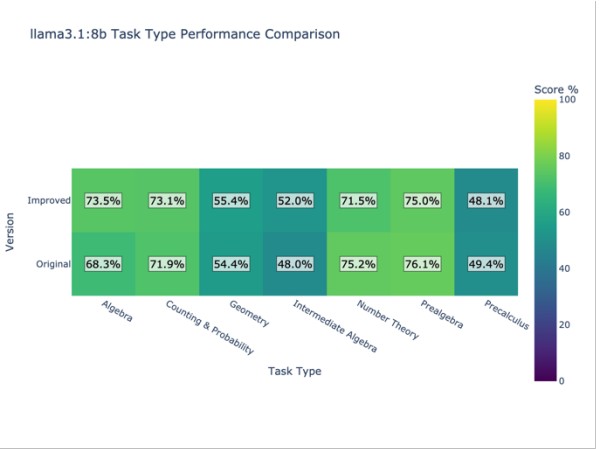

Figure 8: Heatmap Comparison

## 5 KEY OBSERVATIONS

The gpt-4o-mini-2024-07-18 (83.7%) outperformed all other models, while open-codestral-mamba:v0.1 (49.2%) exhibited the lowest accuracy, highlighting substantial variability in problem-solving capabilities across architectures.

All models demonstrated a consistent inverse relationship between accuracy and problem difficulty, with performance dropping by 10-32% from Level 1 to Level 5 tasks.

Algebra emerged as the strongest domain, while Number Theory posed the greatest challenge, suggesting domain-specific architectural biases. Faster-executing models (e.g., ministral-8b-2410: 0.94s) did not correlate with higher accuracy, indicating computational efficiency does not inherently improve solution quality.

Marginal accuracy gains (+0.8%) for llama3.1:8b were accompanied by a 36.7% reduction in resulted program execution time, implying potential for resource-optimized code generation despite limited problem-solving improvements.

<1% of generated code contained unsafe constructs (e.g., infinite loops), necessitating stricter runtime sandboxing.

3.17% of problems remained unsolved by all models after 10 attempts (the mark is $\leq 3$), underscoring the need for enhanced reasoning techniques.

## 6 SUMMARY AND CONCLUSIONS

This study evaluated 10 language models on 945 competition-level mathematical problems, revealing critical insights (24) into their problem-solving capabilities. Commercial models (e.g., gpt-4o-mini) significantly outperformed open-source counterparts, with a 34.5% accuracy gap between the best and worst performers. Domain-specific weaknesses, particularly in Number Theory, persist across architectures.

While iterative regeneration provided minimal accuracy improvements for llama3.1:8b, its 36.7% faster execution time of generated code suggests utility in latency-sensitive applications (25). The technique's domain-specific efficacy warrants further investigation. Future Directions:

- Exploration of RAG frameworks (or similar) to address unsolved problems.
- Domain-specific fine-tuning to bridge performance gaps in challenging areas like Number Theory.

These findings (24) establish a benchmark for LLM-based mathematical problem-solving while identifying actionable pathways for improving accuracy, and computational efficiency.

## REFERENCES

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
