# OpenReview forum: "Token-by-Token Regeneration and Domain Biases: A Benchmark of LLMs on Advanced Mathematical Problem-Solving"
_mathai.club/MathAI/2025/Conference — MathAI 2025 Oral_

### Official Review · Reviewer_AdHR · 2025-02-26
**Rather a technical report**

**Rating:** 3
**Confidence:** 4

**Review:**

The paper has many technical and logical flaws. I cannot seriously evaluate it, it is just a bunch of graphs without any details and background.

**Strengths:**

- Topic is actual LLMs really needed a comprehensive set of benchmarks.
- Mathematical problems are still not solved so well by LLMs without using side tools like CAS and others, so this aspect could be improved.

**Weaknesses:**
- The paper is deanonymized, but as we agreed, this is not a reason for desk rejection.
- The conference template is not used
- The quality of the images is poor. For double columns I would rather use .eps format, so they can be zoomed in.
- Authors' contribution is not clear, please state it in the introduction
- Abstract is too long and filled with technical details without any description
- I could not understand how conclusions are related to experiments.

I would like to recommend to use the template, clearly state the contribution, make a 'red line' throughout the paper, starting from the hypothesis in the intro, going through the experiments and supported by the conclusion. Also, I don't want to discuss the content in depth, as the paper lacks basic things and looks like a technical report.

UPD 27.02: Recent paper discusses LIMO approach https://arxiv.org/pdf/2502.03387 could be used as reference of a good paper on this topic. Moreover, closer experimental setup is used.

---

### Official Review · Reviewer_6bEQ · 2025-02-26
**Poor results analysis and research approach**

**Rating:** 5
**Confidence:** 5

**Review:**

The paper presents a benchmarking study of 10 large language models (LLMs) on competition-level mathematical problems, focusing on their ability to generate executable Python code. While the study attempts to address inconsistencies in mathematical notation and explores token-by-token regeneration as a refinement technique, several critical issues undermine its quality and relevance.

1) Anonymity Violations:
The paper fails to maintain anonymity, as it includes identifiable information such as the author's name, affiliation, and email address. This is a significant breach of the double-blind review process.

2) Format Inconsistencies:
Uses other template in all aspects.

3) Limited Research Value:
The study relies on an existing benchmark (MATH dataset) and evaluates a narrow selection of models (7B–8B parameters) without clear justification for their inclusion. The choice of models lacks motivation, for example reseacrh can compare different types of models with there strong sides:
code generation,
mathematical reasoning (Qwen) ,
tools usage (Mistral series),
etc
And so, on the other hand, the study does not contain a comparison of the proposed models, their training data, techniques, architectures, etc., it is not considered why such results of their comparison were obtained.Given the rapid evolution of LLMs, the findings are likely to lose relevance by the time of the conference, as the results primarily compare specific versions of models with predictable outcomes, such as performance degradation with increasing problem complexity.

Superficial Treatment of Token-by-Token Regeneration: The proposed method for improving model efficiency is only superficially explored, limited to a single architecture (llama3.1:8b). The marginal accuracy gains (+0.8%) and reduced execution time (36.7%) are not thoroughly analyzed, nor are the broader implications for other models or domains.

Conclusion: The paper is not recommended for publication in its current form. The lack of anonymity, formatting issues, limited research value, and superficial treatment of the proposed method significantly detract from its contribution. The study would benefit from a more rigorous approach, including a broader and more motivated selection of models, deeper analysis of the proposed method, and a focus on long-term relevance in a rapidly evolving field.



Update:
Some improvments in template.

---

### Official Review · Reviewer_q2CM · 2025-02-26
**Benchmark with unclear contribution**

**Rating:** 3
**Confidence:** 4

**Review:**

The paper presents a benchmark of 10 large language models on the MATH dataset, aiming to explore the impact of token-by-token regeneration on mathematical problem-solving.

### Strengths
- The paper addresses a relevant issue in solving mathematical problems, which are among the most challenging tasks for large language models and serve as a critical benchmark for reasoning capabilities.
- The study demonstrates that token-by-token regeneration on the LLaMA model significantly increases inference speed without compromising performance.

### Downsides
- The paper is not anonymized and includes author names, affiliations, and contact information, violating submission guidelines.
- The manuscript does not follow the required template format.
- The benchmark for regeneration was run only once, making the observed score differences less representative. At a minimum, standard deviation measurements for the results would improve reliability.
- It is unclear why only 8B-parameter models were used alongside GPT-4, whose parameter count remains undisclosed.
- No background or justification is provided for the token-by-token regeneration approach.

### Conclusions
The study highlights the potential of token-by-token regeneration for mathematical problem-solving with LLMs. However, the results lack reliability due to the experimental setup, which could be improved. Furthermore, the paper's lack of anonymization and deviation from the required template prevent its acceptance in its current state.

---

### Official Review · Reviewer_spvF · 2025-02-27
**There are problems in the "Token-by-Token Regeneration and Domain Biases: A Benchmark of LLMs on Advanced Mathematical Problem-Solving" paper**

**Rating:** 6
**Confidence:** 3

**Review:**

This paper is devoted to solution of such important task as solving mathematical problems using large language models. Corresponding experiments have been provided.

This paper has the following disadvantages:
1) Absence of mention of well-known hybrid approach to solution of mathematical problems when description of each arithmetic problem is translated by LLM from natural language to set of formulas to be proven by SMT solvers (for example, see the "SATLM: satisfiability-aided language models using declarative prompting" paper).
2) Absence of short description of paper structure in the end of "Introduction" section.

---

### Decision · Program_Chairs · 2025-03-08

**Decision:**

Accept (Oral)

**Comment:**

Your article has been accepted and you can make a presentation on the article. All articles will be sorted by rating and within the available conference places one author from each article will be invited. If there are not enough places, then you will either have the opportunity to present remotely or come at your own expense!